# Interplay between Comorbidities and Long COVID: Challenges and Multidisciplinary Approaches

**DOI:** 10.3390/biom14070835

**Published:** 2024-07-11

**Authors:** Rasha Ashmawy, Esraa Abdellatif Hammouda, Yousra A. El-Maradny, Iman Aboelsaad, Mai Hussein, Vladimir N. Uversky, Elrashdy M. Redwan

**Affiliations:** 1Clinical Research Administration, Directorate of Health Affairs, Ministry of Health and Population, Alexandria 21554, Egypt; mri.rasha.m.informatics19@alexu.edu.eg (R.A.); mri.iman.m.informatics18@alexu.edu.eg (I.A.); mai.mk.hussein@gmail.com (M.H.); 2Biomedical Informatics and Medical Statistics, Medical Research Institute, Alexandria University, Alexandria 21561, Egypt; hiph.eabdellatif@alexu.edu.eg; 3Clinical Research Department, El-Raml Pediatric Hospital, Ministry of Health and Population, Alexandria 21563, Egypt; 4Pharmaceutical and Fermentation Industries Development Center, City of Scientific Research and Technological Applications (SRTA-City), New Borg EL-Arab 21934, Alexandria, Egypt; hiph.ymaradny@alexu.edu.eg; 5Microbiology and Immunology, Faculty of Pharmacy, Arab Academy for Science, Technology and Maritime Transport (AASTMT), El-Alamein Campus, Aswan 51718, Egypt; 6Department of Molecular Medicine and USF Health Byrd Alzheimer’s Research Institute, Morsani College of Medicine, University of South Florida, Tampa, FL 33612, USA; 7Department of Biological Science, Faculty of Science, King Abdulaziz University, Jeddah 21589, Saudi Arabia; 8Therapeutic and Protective Proteins Laboratory, Protein Research Department, Genetic Engineering and Biotechnology Research Institute, City of Scientific Research and Technological Applications, New Borg EL-Arab 21934, Alexandria, Egypt

**Keywords:** SARS-CoV-2, COVID-19, long COVID, comorbidities, autoantibodies, interplay, synergistic effects

## Abstract

Long COVID, a name often given to the persistent symptoms following acute SARS-CoV-2 infection, poses a multifaceted challenge for health. This review explores the intrinsic relationship between comorbidities and autoimmune responses in shaping the trajectory of long COVID. Autoantibodies have emerged as significant players in COVID-19 pathophysiology, with implications for disease severity and progression. Studies show immune dysregulation persisting months after infection, marked by activated innate immune cells and high cytokine levels. The presence of autoantibodies against various autoantigens suggests their potential as comorbid factors in long COVID. Additionally, the formation of immune complexes may lead to severe disease progression, highlighting the urgency for early detection and intervention. Furthermore, long COVID is highly linked to cardiovascular complications and neurological symptoms, posing challenges in diagnosis and management. Multidisciplinary approaches, including vaccination, tailored rehabilitation, and pharmacological interventions, are used for mitigating long COVID’s burden. However, numerous challenges persist, from evolving diagnostic criteria to addressing the psychosocial impact and predicting disease outcomes. Leveraging AI-based applications holds promise in enhancing patient management and improving our understanding of long COVID. As research continues to unfold, unravelling the complexities of long COVID remains paramount for effective intervention and patient care.

## 1. Introduction

Long COVID is an umbrella term referring to the collection of signs, symptoms, and conditions persisting or developing after the acute phase of COVID-19 infection caused by severe acute respiratory syndrome coronavirus-2 (SARS-CoV-2) [1]. The virus emerged in late 2019 in Wuhan, China, and rapidly disseminated, leading to a pandemic that posed an unprecedented burden to health systems globally [2]. While research has heavily focused on the acute phase of the disease, emerging evidence highlighted the complex and debilitating condition known as long COVID [1,3]. With symptoms persisting for weeks to months or even years after the acute infection resolves, coupled with a high prevalence of up to 50% among COVID-19 survivors [4] and numbers escalating by the day, long-term intensifies the public health burden of the illness [1].

Long COVID is the most common term coined by patients, though other terminologies have been used in the literature, such as long-term effects of COVID, post-COVID syndrome, long-haul COVID, post-acute COVID, chronic COVID, or post-acute sequelae of SARS-CoV-2 infection (PASC) [5]. Long COVID is currently an active area of research due to the many uncertainties around the condition. To this day, it lacks a unified timeframe for diagnosis. While the World Health Organization adopted a long COVID definition as a health condition where symptoms persist or arise within three months from initial infection, the Center of Disease Control and Prevention (CDC) and other agencies settled for at least four weeks from the infection onset [5,6]. For now, specific diagnostic tools are still being developed, and diagnosing the condition is challenging, mostly relying on symptoms and excluding potential alternative diagnoses from suspected or confirmed COVID-19 cases [6]. Discrepancies in definition have led to significant differences in estimates of prevalence and incidence, in addition to the timeframe and population studied. The incidence was 10–30% and 50–70% in non-hospitalized and hospitalized cases, respectively [1,7,8]. A lower incidence of 10–12% was observed in vaccinated cases [1,9,10].

Long COVID symptoms vary widely, with a growing list of more than 200 signs and symptoms encompassing multiple systems that may take a remitting–relapsing pattern [1,11]. Symptoms like dyspnea, myalgia, anosmia and dysgeusia, sleep disorders, and memory problems are commonly reported [12,13]. Post-exertional malaise occurs when mental or physical exertion triggers or worsens symptoms [14]. Long COVID afflicts respiratory, cardiovascular, neurological, gastrointestinal, and psychological functions [1] and induces diabetes [15], immune system dysregulation [16], and multiple organ injuries. Respiratory repercussions may scale up to pulmonary fibrosis. Cardiovascular sequelae include thromboembolic events, deep venous thrombosis, and myocarditis [17]. Neurological symptoms involve headaches, neuropathic pain, and cognitive impairment [18]. Psychological effects involve depression, anxiety, and post-traumatic stress disorder [19]. Consequently, patients enduring long COVID suffer hampered ability to carry out their everyday functions and have lower quality of life, and those with severe symptoms may require specialized hospital care.

The etiology of long COVID still needs to be fully understood. Given the diversity of symptoms and phenotypes, several mechanistic studies are underway testing the multiple suggested hypotheses [1,20]. However, long COVID was mostly likely associated with female sex, older age, lower socioeconomic status, and underprivileged communities or minor ethnic groups [21,22]. Smokers and those with pre-existing clinical conditions, such as obesity, diabetes, cardiovascular disease, neurological conditions, immunodeficiency, genetic predisposition, COVID-19 vaccination status [13], and the type of variant plays a crucial role. A higher number of symptoms, higher severity of the acute COVID illness, and associated prolonged hospitalization also increased the probability of experiencing long-term effects. Current evidence suggests that several overlapping mechanisms, rather than a single cause, instigate long COVID [1,11].

The relationship of chronic health conditions with long COVID may be bidirectional. As mentioned above, long COVID prevails mostly in the presence of chronic comorbidities which in turn instigate the development of chronic sequelae. Comorbidities impair the body’s ability to combat the infection. The molecular signature of comorbidities could explain long COVID since the heightened inflammatory milieu contributes to higher oxidative stress, sustained tissue damage, and organ dysfunction, thus exacerbating the long COVID symptoms and extending its duration. Immune system dysregulation represented by enhanced cytokine signaling, overproduction of interleukin-1 (IL-1), interleukin-6 (IL-6), tumor necrosis factor-alpha (TNF-α), and interferon (IFN)-γ leads to the persistent inflammatory responses [23]. Additionally, endothelial dysfunctions in long COVID influence blood flow, increasing vascular permeability and activating the thrombogenesis cascade. Similarly, SARS-CoV-2 could penetrate the CNS and induce neurological sequelae, aggravating the condition [1,24]. Generally, recovery patterns differ by individual characteristics, the severity of symptoms, the onset of treatment, and the organ involved [25]. Almost 60% of long COVID symptoms emerge at week 4 and are resolved by week 12. Beyond week 12, slow recovery or a plateau is observed [26].

In light of our previous articles [27,28,29,30] to conceptualize the reasons behind long COVID, the current review will explore the interplay between comorbidities and long COVID to provide a comprehensive understanding and elucidate their synergistic effects on disease progression and resolution.

## 2. Long COVID and Autoantibody

### 2.1. Unraveling Persistent Symptoms and Immune Responses

The global impact of the COVID-19 pandemic on health, economies, and social norms has created a new reality for individuals worldwide. As societies grapple with the complexities of this ongoing crisis, understanding its long-term implications, such as the diverse array of persistent symptoms in individuals with “long COVID,” is crucial [1]. Autoantibodies have emerged as a compelling area of research in understanding the pathophysiology of COVID-19, shedding light on the potential role of dysregulated immune responses in disease progression. These autoantibodies have been implicated in the development of severe COVID-19 complications and may serve as a key mechanism driving immune-mediated damage. Studies have reported a 42.6% higher likelihood of developing autoimmune complications in COVID-19 patients compared to matched nonpatients [31]. Furthermore, the presence of autoantibodies in COVID-19 patients has been associated with increased disease severity, highlighting their significance as a potential comorbid and complicating factor in the management of this complex disease [32].

Although the pathophysiology of long COVID is incompletely understood, several theories have been posited to explain the molecular root of immune dysregulation, including DNA replication by viral proteins, systemic manifestation, and multiorgan involvement of COVID-19 due to broad expression of the SARS-CoV-2 receptor angiotensin-converting enzyme 2 (ACE2), activation of immune cells, release of autoantigens from infected tissue, superantigen-mediated activation of lymphocytes, and epitope spreading.

### 2.2. Molecular Roots and Immune Dysregulation in Long COVID: Insights from Studies

Studies examining immune dysregulation in individuals with long COVID revealed highly activated innate immune cells, a death of naive T and B cells, and elevated expression of type I and type III interferons (interferon-β (IFNβ) and IFNλ1), persisting for at least 8 months [33]. Additionally, exhausted T cells, diminished numbers of CD4+ and CD8+ effector memory cells, and elevated PD1 expression on central memory cells persist for at least one year after infection. The expansion of cytotoxic T cells has been associated with the gastrointestinal presentation of long COVID [21]. Other studies have found elevated levels of cytokines, particularly interleukin-6 (IL-6), IL-1β, tumor necrosis factor-alpha (TNF-α), and interferon gamma-induced protein 10 (IP-10). Patients with long COVID demonstrated elevated levels of autoantibodies to ACE2, β2-adrenoceptor, muscarinic M2 receptor, angiotensin II AT1 receptor, and angiotensin 1–7 MAS receptor [34,35]. Other studies proved that low antibody count during the acute phase of COVID-19 is a predictor for long COVID [36,37]. In addition, SARS-CoV-2 infection has been associated with various autoimmune diseases. The immune system’s main affected systems lead to broad autoimmune diseases, especially rheumatic manifestations [35,38,39], followed by the circulatory system, resulting in the elevated risk of deep vein thrombosis and pulmonary embolism [38,40,41]. 

Recent studies revealed an association between several specific autoantibodies and long COVID. The most prevalent antibodies are antibodies to small nuclear ribonucleoprotein (U1-snRNP), anti-SSB/La and anti-SUMO1-DHX35 autoantibodies, autoantibodies against G-Protein Coupled Receptors (GPCRs), anti-cardiolipin antibodies (aCLs), and anti-neutrophil cytoplasmic antibodies (ANCAs) [35,42,43,44]. The molecular mechanisms underlying the function of these antibodies remain an active area of ongoing research, warranting further investigation and analysis. However, emerging evidence has begun to demonstrate potential physio-pathological mechanisms for certain autoantibodies. For instance, U1-snRNP was found to be responsible for persistent fatigue and dyspnea (*p* = 0.02) and led to the emergence of post-COVID systemic lupus erythematosus (SLE) by activating the nucleotide-binding domain, leucine-rich-containing family, and pyrin domain-containing-3 (NLRP3) inflammasome and inducing IL-1β production [45]. Autoantibodies against SUMO1-DHX35 were also prevalent in 7% of long COVID female patients despite its potential pathogenicity mechanism still to be elucidated [44].

### 2.3. Immune Complexes as Potential Markers for Critical COVID-19 Disease Progression

Recent studies suggest that an increased inflammatory response and formation of IgG immune complexes (ICs) may significantly contribute to severe and prolonged COVID-19 disease progression [46]. Deposits of unresolved ICs in tissues can trigger an overactive Fc gamma receptor (FcγR)-mediated signaling cascade, which may lead to common IC-associated organ diseases such as vasculitis, glomerulonephritis, and arthritis. Additionally, myeloid cells exhibit enhanced eosinophil-mediated inflammation in the respiratory tract of critically ill COVID-19 patients, which is linked to FcγR signaling [47]. These comorbidities have been frequently reported in long COVID patients. Moreover, studies have found that soluble ICs (sICs) are also present in the circulation of most severely ill patients, and their systemic abundance correlates with disease severity [48]. Therefore, detecting circulating sICs in patients may be a potential marker for critical COVID-19 disease progression. Early detection of sICs after clinical deterioration may indicate the need for prompt anti-inflammatory treatment [49]. The immunological and autoantibody response is presented in Figure 1.

### 2.4. Autoantibodies in COVID-19-Associated Thrombosis

SARS-CoV-2 could infect the vascular endothelial cells, leading to a blockage of small blood vessels by inflammatory cells and thrombi in large blood vessels. This can result in endothelial hyperplasia. Additionally, SARS-CoV-2 can directly cause damage and apoptosis and reduce the normal endothelium’s antithrombotic activity. This reduction in antithrombotic activity is characterized by increased levels of von Willebrand factor (vWF), fibrinogen, and factor VIII in COVID-19 patients. However, patients infected with SARS-CoV-2 have been reported to have elevated levels of IL-6, IL-1β, IFN-γ, MCP-1, MIP, and IP10. Pro-inflammatory cytokines may interfere with the endothelial function and integrity, leading to the release of vWF. Additionally, there may be an upregulation of adhesion molecules, such as intercellular adhesion molecule 1 (ICAM-1), vascular endothelial growth factor (VEGF), αvβ3 integrins, and P- and E-selectins. Moreover, endothelial cytokines and chemokines may also be produced [50,51]. Emerging research posits that thrombosis in the context of antiphospholipid syndrome (APS) is multifaceted and intricately linked to the presence of antiphospholipid autoantibodies (aPLs) [52]. These antibodies are pivotal in promoting thrombosis by activating endothelial cells and platelets and stimulating neutrophils to release neutrophil extracellular traps (NETs). Patients hospitalized with COVID-19 have elevated levels of NETs, which correlate with disease severity and thrombosis [53].

### 2.5. Autoimmune Reactions and Connective Tissue Diseases

The most commonly reported long COVID symptoms are connective tissue diseases including inflammatory arthritis [54], myalgia [55,56], rheumatoid arthritis (RA), and SLE. Most of the patients were found to have one of the following autoantibodies, suggesting a post-COVID-19 autoimmune reaction: antineutrophil cytoplasmic antibodies [57], anti-cyclic citrullinated peptide (anti-CCP) with a majority of the antinuclear antibody (ANA), aCL, and anti-beta-2-glycoprotein-1 (anti-β2GP1) in association with a cytokine storm [58].

The duration of autoantibody presence in long COVID is not thoroughly documented. The formation of these autoantibodies seems to be a reactionary phenomenon, implying their transient nature and eventual disappearance in COVID-19 individuals. However, emerging evidence indicates a potential association between acute COVID-19 infection and the development of autoimmune connective tissue diseases [56,59].

## 3. Cardiovascular Complications

### 3.1. In-Depth Examination of Cardiovascular Issues Associated with Long COVID

The cardiovascular complications associated with long COVID have garnered increasing attention in the medical community due to their significant impact on individuals recovering from the initial viral infection [60]. Multiple cardiac and extracardiac pathological consequences, such as residual respiratory disorders with significantly lower peak-of-maximum oxygen consumption, pulmonary hypertension, muscular deconditioning, cytokine dysregulation, left or right ventricular dysfunction, chronotropic incompetence, altered parasympathetic tone, or increased heart rate variability, contribute to the cardiovascular symptoms of long COVID [57]. Studies have documented a range of cardiovascular issues in patients with long COVID, including myocardial inflammation, arrhythmias, myocardial infarction, dysautonomia, vascular thrombosis, and heart failure [60,61]. Additionally, emerging evidence suggests that individuals with pre-existing cardiovascular conditions may be at higher risk for developing severe cardiovascular complications during the extended recovery phase of COVID-19 [62]. Figure 2 provides a detailed illustration of the complications, diagnosis, and treatment of long COVID comorbidities.

### 3.2. Discussion of the Potential Mechanisms Involved

The proposed mechanisms for COVID-19’s impact on the cardiovascular system are still under investigation. One hypothesis attributed it to the direct effects of the SARS-CoV-2 on the cardiovascular system and the inflammation associated with long COVID [63]. Another one suggests that autoimmunity to cardiac self-antigens could be triggered through “molecular mimicry”, in which antigens shared between COVID-19 and host cells cause the immune system to attack the body’s tissues, or through “bystander loss-of-tolerance”, in which T or B cells with random self-reactive specificities are accidentally activated after receiving co-stimulation signals from the infection [64]. This hypothesis could explain many features of acute and long COVID, including the variation in T and B cell antigen specificities among individuals, the long-lasting and independent nature of autoimmune responses, and the age-dependent increase in inflammation that could lead to the development of long COVID. 

In a hyperinflammatory state, oxidative phosphorylation may be active. Viral infections are thought to cause a shift in the mitochondrial energy system involvement from ATP synthesis to innate immune signaling, which occurs to eliminate pathogens, promote inflammation, and eventually restore tissue homeostasis [65]. Glycolysis increases and oxidative phosphorylation decreases. Oxidative stress has been identified in many acquired myocardial disorders and could trigger substantial autonomic dysfunction [66,67].

### 3.3. Addressing Potential Interactions between Different Comorbidities and Their Impact on Long COVID

Although long COVID has been diagnosed in men and women of all ages, the most serious cases and deaths have only been reported in a subset of people. Several longitudinal studies conducted in Europe and Asia found an increase in cardiac events in long COVID patients who had hypertension, a high lipid profile, and coronary artery disease [68,69]. However, most studies in the literature had a short follow-up period and a restricted selection of cardiovascular outcomes, and they were all restricted to hospitalized individuals (who make up the minority of COVID-19 cases) [51,70].

## 4. Respiratory Complications

Long COVID frequently includes persistent respiratory symptoms lasting well beyond acute infection. Understanding these symptoms requires differentiating between airway and parenchymal diseases. Airway diseases may present as obstructive conditions (like asthma) or combined obstructive–restrictive patterns, similar to chronic obstructive pulmonary disease (COPD). Parenchymal diseases can cause restrictive lung patterns, with imaging revealing interstitial lung disease, such as lymphocytic interstitial pneumonitis [71]. Even a year after hospitalization, approximately one-third of COVID-19 pneumonia patients exhibit persistent chest CT abnormalities, including fibrosis, air trapping, and bronchiectasis. Some patients also face a sustained higher risk of venous thromboembolic disease. Late-stage COVID-19 histopathology shares similarities with other acute lung injuries, demonstrating organizing and chronic fibrosing patterns, reminiscent of the SARS epidemic [72].

### 4.1. Analysis of Persistent Respiratory Symptoms

In a study by the European Respiratory Society, it was noted that COVID-19 can lead to persistent respiratory symptoms, even in mild cases. Many patients, including those with mild symptoms, continue to experience issues like cough and difficulty breathing after recovering from the virus. The study suggests that while COVID-19 primarily affects the respiratory system, the exact mechanisms behind these persistent symptoms are not yet fully understood [73]. Various patient-based studies found that coughing was prevalent in over 50% of cases, and dyspnea (difficulty breathing) was reported in over 40% of patients [74]. The study indicates that medical imaging suggests potential ongoing respiratory inflammation or damage, but this remains mostly theoretical. One possible explanation highlighted is the presence of partially treated or undertreated pulmonary venous thromboembolism, leading to associated pulmonary inflammation and fibrosis [75,76]. The findings are supported by discharge material analysis from New York hospitals, where approximately 88% of patients still showed some level of inspiratory pathological evidence of COVID-19 [77,78]. This suggests that there might be residual lung abnormalities and breathlessness even after recovery. While more research is needed, these findings suggest a need for further investigation into the long-term respiratory effects of COVID-19 and potential treatment strategies. Long COVID can lead to persistent respiratory symptoms such as cough, breathlessness, and fatigue. Some studies suggest that these symptoms may be related to ongoing inflammation, lung damage, or other underlying mechanisms [79].

### 4.2. Autoimmune Antibodies and Lung Involvement

The development of autoimmune antibodies following COVID-19 infection has emerged as a potential driver of symptoms in long COVID. Studies have detected various autoantibodies, including antinuclear antibodies (ANAs), anti-phospholipid antibodies, and tissue-specific autoantibodies in patients with lingering COVID-19 complications [51,80]. Interestingly, autoantibodies targeting lung tissue have been implicated in certain autoimmune lung diseases, suggesting a possible link between autoimmunity and respiratory manifestations in long COVID [81]. However, more research is warranted to establish a causal relationship between autoimmune antibodies, lung function abnormalities, and persistent respiratory symptoms after COVID-19.

### 4.3. Lung Function Abnormalities in Long COVID

Various issues with breathing among patients reveal a need for comprehensive research on persistent respiratory symptoms lasting over twelve weeks. Long COVID, impacting both physical and mental health, is a complex condition resulting from various infections [82]. Long COVID, the constellation of lingering symptoms persisting after initial SARS-CoV-2 infection, often includes respiratory dysfunction. Studies have observed decreased lung diffusion capacity, restrictive lung patterns, and small airway abnormalities in long COVID patients, suggesting potential long-term pulmonary damage [83]. While the mechanisms underlying these abnormalities are complex, ongoing inflammation, residual fibrosis, and vascular changes have been proposed as contributing factors [84]. Further research is crucial to delineate the precise nature and duration of these respiratory sequelae in long COVID patients.

## 5. Neurological Complications

### 5.1. Long COVID’s Diverse and Systemic Neurological Impact

Long COVID can manifest in a wide range of neurological symptoms, extending beyond the typical respiratory issues. This complexity underscores the need for comprehensive understanding for effective clinical management [84,85]. The spectrum of reported symptoms encompasses mild cognitive issues and headaches to more severe conditions like neuropathic pain and seizures [84]. Notably, studies suggest an increased risk of cerebrovascular events and encephalopathies, highlighting the systemic nature of long COVID’s neurological involvement [86]. While further research with larger, confirmed COVID-19 cohorts is necessary, the reported symptoms in this article align with recent findings [87]. Notably, neurological symptoms can be the initial presentation, even in mild COVID cases, with prevalence estimates ranging from 70 to 90% (headaches affecting up to 60%), alongside dizziness and sleep disturbances. Additionally, more pronounced symptoms like disorientation, cognitive impairment, and “brain fog” can occur in both mild and severe COVID cases [87].

### 5.2. Long COVID’s Potential Neurological Mechanisms

Long COVID can manifest in various ways, often including neurological complications. Understanding the potential mechanisms underlying these neurological issues is crucial for developing targeted treatments. One potential contributor is persistent inflammation [86]. This sustained inflammatory state can trigger a cascade of events leading to neurological damage. The blood–brain barrier (BBB), a highly selective membrane protecting the brain from harmful substances, may be compromised in long COVID. This disruption, potentially caused by elevated levels of inflammatory molecules like interleukin-6 (IL-6), could allow harmful substances or immune cells to infiltrate the brain, causing damage and dysfunction [88]. Furthermore, the immune system’s response to the virus itself might be a contributing factor. In some cases, the body’s defense mechanisms can become misdirected, leading to an autoimmune response that mistakenly attacks healthy tissues within the nervous system. This can explain the array of long-term neurological symptoms experienced by some individuals after recovering from the initial COVID-19 infection [89,90].

Another potential mechanism is viral persistence, where the SARS-CoV-2 might remain dormant within the nervous system. This persistence could either directly damage nearby neurons or trigger an ongoing immune response, contributing to chronic inflammation and associated neurological issues [87]. The combined effects of these potential mechanisms, including neuroinflammation, neuronal injury, and degeneration, may ultimately lead to the diverse neurological symptoms observed in long COVID patients. While the exact mechanisms and individual risk factors remain under investigation, ongoing research holds promise for a clearer understanding, and ultimately the development of effective treatments for managing long COVID’s complex neurological manifestations [91].

### 5.3. Neurological Symptoms in Long COVID

Beyond the well-known respiratory and cardiovascular concerns of long COVID, a growing body of research reveals a concerning prevalence of neurological symptoms in this population. One prominent symptom is headache, often described by long COVID patients as distinct from typical migraines or tension headaches. These headaches tend to be more severe, often accompanied by neck tension, and demonstrate limited response to conventional pain medication. Research suggests this may be due to complex mechanisms involving inflammation and central sensitization within the nervous system [92]. Dizziness, another frequently reported symptom, presents as light-headedness and can affect individuals of all ages. While the exact cause remains under investigation, potential contributors include inner ear dysfunction, autonomic nervous system disruption, and anxiety. Sleep disturbances, particularly insomnia, significantly impact a large portion of long COVID patients. Studies suggest this may be due to a combination of factors, including the direct effects of the virus on sleep–wake regulatory centers, elevated stress and anxiety levels, and disrupted daily routines [93].

Beyond these prevalent concerns, other neurological symptoms associated with long COVID include cognitive impairment (“brain fog”), fatigue, loss of taste and smell, neuropathic pain, mood disorders, and in rare cases, seizures [94]. The diverse range of these manifestations underscores the complexity of long COVID and the need for further research to elucidate the underlying mechanisms. Additionally, a multi-disciplinary approach to patient care is crucial, encompassing specialists from neurology, rehabilitation, psychology, and other relevant fields, to address these multifaceted challenges.

## 6. Psychosocial Impact

### 6.1. Psychological and Social Consequences of Long COVID

Research showed that anxiety and depression are prevalent psychological sequelae of long COVID, contributing to heightened distress and emotional dysregulation [95]. The psychological impact of long COVID, including anxiety, irritability, and excessive feelings of stress or anger, has been well documented [96]. Studies have shown that the pandemic could have long-lasting psychological effects, leading to mild to severe depression, anxiety, and stress [97]. Additionally, cognitive dysfunction, including memory and concentration difficulties, was commonly reported among long COVID patients, which were posing challenges to work, academic, and social activities [98]. Furthermore, a substantial proportion of individuals with long COVID experienced symptoms consistent with post-traumatic stress disorder (PTSD), such as intrusive thoughts and hypervigilance. These symptoms often stemmed from the traumatic experience of illness and hospitalization [99]. Even more, two years after the COVID-19 outbreak, health workers showed a significant level of depression, anxiety, stress, and remarkable signs of psychological distress [100]. Also, the lives and behaviors of the general population, including perinatal women, have been significantly affected by the uncertainties caused by COVID-19, leading to significant psychological stress [101].

The psychological consequences of long COVID were compared between different groups, such as school-going adolescents, COVID-19 survivors, and their relatives, indicating a widespread impact on mental health [102,103,104,105]. The psychological state of college students was found to be more vulnerable to the outbreak of COVID-19 than other groups [106]. Additionally, the existing literature on the mental health consequences of the pandemic on both the general population and persons with confirmed COVID-19 showed that depression, anxiety, and other mental health problems are common [107]. Moreover, the pandemic has been associated with increased health-related socioeconomic vulnerability among women, which is prevalent and associated with alarmingly high rates of mental health problems [108].

Longitudinal data from the social study at University College London (UCL) on COVID-19 were analyzed to compare trajectories of depressive and anxiety symptoms before and after getting COVID-19 between matched long- and short-COVID groups. It highlighted the long-lasting impact of COVID-19 on mental health [109,110]. Furthermore, the associations of long COVID symptoms, clinical characteristics, and affective psychological constructs in a non-hospitalized cohort were studied, emphasizing the need for understanding the correlations of long COVID symptoms and affective psychological constructs on non-hospitalized patients [111,112].

More studies are needed to understand the pathogenetic mechanisms underlying the neuropsychiatric sequelae of long COVID, identifying potential key targets for developing effective treatment strategies [113]. Therefore, the long-term psychological consequences of COVID-19 on survivors have been highlighted, emphasizing the need for appropriate mental health services to offer psychiatric support [113].

### 6.2. Challenges Faced by Individuals with Prolonged Symptoms

Long COVID, in addition to its psychological impact, can have profound social consequences. It could lead to stigma, social isolation, and disruptions in interpersonal relationships. The stigma surrounding COVID-19 is fueled by misconceptions and fear of contagion. This results in discrimination and social rejection towards individuals with long COVID, exacerbating feelings of shame and self-blame [96]. Moreover, the unpredictable nature of long COVID symptoms, coupled with the lack of understanding and support from others, contributed to social isolation and withdrawal from social activities, further exacerbating feelings of loneliness and disconnection [93]. Social isolation and loneliness were identified as significant outcomes of the COVID-19 pandemic, particularly for individuals with long COVID [114]. Furthermore, disruptions in interpersonal relationships, including strain within families, challenges in maintaining employment, and difficulties in accessing healthcare services, compounded the social burden experienced by individuals with long COVID [115]. Studies indicated that exposure to outbreaks had various psychological effects, which might have long-term consequences and affect decision-making capacity [116].

The psychological and social consequences of long COVID have underscored the critical need for comprehensive healthcare policies to support affected individuals and communities. Strategies aimed at enhancing access to mental health services, promoting public awareness and education, combating stigma, and fostering social support networks are essential to address the multifaceted needs of long COVID patients [117]. Moreover, investment in research infrastructure, including longitudinal studies and clinical trials, is needed to advance our understanding of long COVID and inform evidence-based interventions [118]. Collaborative efforts between healthcare providers, policymakers, researchers, and community organizations are essential to develop holistic approaches to mitigate the long-term impact of COVID-19 on mental health and social well-being [119]. Furthermore, a previous study emphasized the importance of promoting equity in COVID-19 treatment, particularly considering the disproportionate impact of the pandemic on marginalized groups [119]. This highlights the need for healthcare policies that address social disparities and ensure equitable access to healthcare services for long COVID patients.

## 7. Obesity and Its Impact

### 7.1. Influence of Obesity on Inflammatory Mediators and Long COVID Consequences

Several studies highlighted the association between obesity and adverse COVID-19 outcomes. For instance, a systematic review and meta-analysis demonstrated that obesity was associated with increased hospitalization and intensive care unit (ICU) admission rates among COVID-19 patients [120]. Furthermore, it was found that obesity is an independent risk factor for severe respiratory diseases, contributing to high mechanical ventilation rates, high mortality, and slow recovery in COVID-19 patients [121].

The underlying mechanisms of the association between obesity and COVID-19 outcomes were related to mechanical and immunologic factors [122]. A previous review highlighted that obesity-dependent circumstances triggered an increased risk for COVID-19 severity and its clinical symptoms. This emphasized the role of obesity as an adipose tissue dysfunction disease and a risk factor for infections, with COVID-19 as a case study [123]. Furthermore, the influence of obesity on inflammatory mediators and long COVID consequences has also been explored. Obesity is associated with chronic low-grade inflammation, characterized by elevated levels of pro-inflammatory cytokines, adipokines, and immune cell activation [124]. This state of low-grade systemic inflammation, or ‘metaflammation’, is a hallmark of obesity and is linked to its co-morbidities [124,125]. Following SARS-CoV-2 infection, obese individuals may experience dysregulated immune responses, including excessive production of pro-inflammatory cytokines such as IL-6, TNF-α, and IL-1β [124]. These heightened inflammatory responses contribute to the pathogenesis of acute respiratory distress syndrome [48], multi-organ dysfunction, and thrombo-inflammatory complications observed in severe COVID-19 cases among individuals with obesity [126,127].

Additionally, obesity was identified as a risk factor for long COVID, with individuals with obesity being at a greater risk for persistent symptoms after COVID-19 recovery. Obesity-related inflammation may play a role in the development and persistence of long COVID symptoms [128]. A longitudinal study reported a higher prevalence of persistent symptoms, including fatigue, dyspnea, cognitive dysfunction, and cardiovascular complications among individuals with obesity following acute COVID-19 infection [127]. Also, a study indicated that obesity was associated with abnormal hyper-ventilatory response and impaired gas exchange during exercise in individuals with long COVID, suggesting a potential link between obesity and long-term respiratory complications [129].

Furthermore, the dysregulated immune response and systemic inflammation associated with obesity may contribute to the perpetuation of symptoms and delayed recovery in these individuals [127]. Also, the dysfunction of adipose tissue induced by chronic over-nutrition and obesity results in dysregulation of both innate and adaptive immunity, leading to local chronic low-grade inflammation, which is characteristic of obesity [130]. Additionally, obesity is linked with conditions associated with immune dysfunction, such as increased susceptibility to infection or bacteremia [125].

Healthcare providers should be vigilant for the development of long COVID symptoms in individuals with obesity and consider tailored approaches to monitoring, diagnosis, and management. Targeted interventions aimed at reducing obesity-related inflammation, such as lifestyle modifications, pharmacotherapy, and bariatric surgery, may help mitigate the risk of long COVID and improve outcomes in this population. Public health strategies focused on obesity prevention and management are critical for reducing the burden of long COVID and its associated complications on healthcare systems and society at large.

### 7.2. Effects of Metabolic Syndrome on Health in the Context of Long COVID

Metabolic syndrome is a complex condition associated with dysregulation of the immune system, characterized by chronic low-grade inflammation, impaired innate and adaptive immune responses, and altered cytokine profiles [131]. These immune dysregulations may contribute to the pathogenesis of long COVID by exacerbating inflammatory responses, impairing viral clearance, and promoting tissue damage and organ dysfunction [131]. Furthermore, metabolic syndrome-associated comorbidities, such as obesity and diabetes, further amplify immune dysregulation and increase susceptibility to severe COVID-19 outcomes. Individuals with metabolic syndrome were predisposed to an increased risk of cardiovascular complications, thromboembolic events, and microvascular dysfunction, which might exacerbate the severity and duration of symptoms in individuals with long COVID [132]. Endothelial dysfunction, hypercoagulability, and systemic inflammation contributed to the development of thrombotic complications and multi-organ dysfunction in individuals with metabolic syndrome and long COVID [133]. 

Additionally, metabolic syndrome-associated cardiometabolic abnormalities, including dyslipidemia, insulin resistance, and hypertension, further contributed to the pathophysiology of long COVID complications, such as myocardial injury, stroke, and renal impairment [134]. Metabolic comorbidities, when combined with metabolic syndrome, are characterized by a low-grade systemic inflammation, which might contribute to the severity of COVID-19 outcomes [135]. Finally, patients with metabolic syndrome hospitalized with COVID-19 pneumonia had significantly higher mortality and intensive care requirements, indicating the impact of metabolic syndrome on COVID-19 outcomes [136].

The effective management of long COVID in individuals with metabolic syndrome requires a multifaceted approach that integrates lifestyle modifications, pharmacological interventions, and rehabilitation programs. Lifestyle modifications, including dietary interventions, physical activity, and weight management, play a pivotal role in improving metabolic health and reducing the risk of long COVID complications. Long-term lifestyle modification strategies are necessary, and behavioral procedures represent the most effective nonsurgical approach [137]. Additionally, the implementation of healthy lifestyles at the community scale is essential in managing metabolic syndrome and its components [138]. A structured program with rehabilitation and physical activity, as well as optimal dietary management, is of utmost importance, especially for patients with metabolic diseases and/or long COVID [139]. Also, pharmacological interventions targeting metabolic syndrome components, such as statins, antihypertensive agents, and antidiabetic medications, may be beneficial in managing cardiovascular risk and mitigating the adverse effects of metabolic syndrome on long COVID outcomes. 

## 8. Diagnostic Challenges 

### 8.1. Analysis of Difficulties in Diagnosing and Characterizing Long COVID

Developing a diagnostic test for COVID-19 is the second research priority of the National Research Institute [140]. Medical professionals typically adhere to a standard diagnostic procedure for identifying long COVID, involving the assessment of symptoms and the exclusion of other conditions to reach a diagnosis. However, there is currently a lack of standardized guidelines specifically designed for the diagnosis of long COVID, which hinders consistent and timely detection [6]. Despite the widespread use of this general diagnostic approach, diagnosis of long COVID presents challenges due to its varied manifestations and diverse clinical presentations, compounded by the absence of uniform definitions and diagnostic criteria. Other factors could also delay the diagnosis of long COVID such as the SARS-CoV-2 variant, patient comorbidities, patient disabilities, and mental health and income level [141]. 

The necessity of an active acute phase, with quantitative polymerase chain reaction (qPCR) positivity, before a long COVID diagnosis is debated, with earlier guidelines requiring a positive laboratory COVID-19 diagnosis, but current guidelines suggest that exposure to the virus, even when it is asymptomatic, can lead to long COVID, posing challenges for patients without initial qPCR or lab-based COVID-19 diagnosis records. A delayed diagnosis of long COVID can lead to adverse consequences at various levels, including individual, community, national, and international ramifications. Clinicians should exhibit caution in diagnosing long COVID due to limited treatment options and potential implications for social welfare benefits. Therefore, there is a need for ongoing discussion regarding the evolving diagnostic criteria and methodologies [141,142].

Various symptoms and risk factors contribute to the definition of long COVID. These symptoms encompass a wide range of manifestations, including chest pain, fatigue, difficulty breathing, post-exertional symptoms, sensory abnormalities, and altered occupational functioning, along with complications affecting cardiovascular, respiratory, nervous system, cognitive, mental health, and physical well-being [143,144]. Additionally, prevalent risk factors identified in these investigations include advanced age (over 65 years), female gender, the presence of comorbidities, and indicators of the severity of acute infection, such as hospitalization and the requirement for supplemental oxygen [145].

### 8.2. Discussion on the Evolving Diagnostic Criteria and Methodologies

As previously mentioned, there is no specific diagnostic test available to identify long COVID, and diagnosis typically relies on a subjective process of exclusion, with separate tests applied to address individual patient complaints. This subjectivity in diagnostic criteria contributes to underestimation, particularly among certain demographic groups. Despite experiencing higher rates of COVID-19 during the acute phase, the black and Hispanic communities exhibit long COVID prevalence rates comparable to those of the Caucasian community. Diagnosis of long COVID poses challenges among elderly individuals or those with comorbidities, as symptoms may exacerbate existing conditions [146]. The episodic nature of long COVID, characterized by fluctuating symptoms over extended periods, hampers daily functioning [143,147]. Raveendran proposed diagnostic criteria for long COVID, encompassing both symptomatic and asymptomatic cases, which relied on identifying individuals with COVID-19, clinical criteria, and disease duration [145].

In line with the diagnostic approaches discussed by Kim et al. [143], the proposed diagnostic tests associated with clinical symptoms of long COVID are summarized below. The pulmonary function test serves as a straightforward and non-invasive diagnostic procedure, recommended after approximately three months for long COVID diagnosis, particularly for patients with severe or critical illness, persistent dyspnea, or pre-existing lung conditions [148]. Persistent respiratory symptoms warrant consideration of a chest X-ray to rule out alternative conditions and detect early signs of lung fibrosis. Chest computed tomography [149] scans should be considered in cases of abnormal chest X-ray findings or persistent symptoms [149]. Transthoracic echocardiography may aid in assessing pericarditis/myocarditis or heart failure in patients with suggestive symptoms occurring over 12 weeks post-acute phase. Functional evaluations, such as the 6 min walking test or 15–30 s sit-to-stand test, are valuable for assessing cardiopulmonary function during patient evaluation or rehabilitation. Different blood tests may be considered based on the symptoms such as C-reactive protein (CRP), erythrocyte sedimentation rate (ESR), complete blood count (CBC), liver function test (LFT), IL-6, and D-dimer [150,151]. 

Additionally, other biomarkers were correlated with increased in-hospital mortality and mortality within one year among COVID-19 patients, such as elevated levels of N-terminal brain-type natriuretic peptide (NT-proBNP) and high-sensitivity cardiac troponin I (hs-TnI). This suggests that NT-proBNP and hs-TnI could serve as valuable prognostic indicators for monitoring individuals with long COVID [152]. Furthermore, another study indicated that suppression of tumorgenity-2 and hs-TnI levels was linked to one-year mortality, although not with in-hospital mortality [153]. Other biomarkers such as exosomes and their cargo including different micro-RNAs were linked with neuroinflammation and cognitive disabilities, including miR-146a, miR-155, miR Let-7b, miR-31, miR-16, and miR21 [27,28,154].

Fatigue severity can be assessed using a fatigue severity scale, while patients with persistent arthralgia or myalgia lasting beyond 12 weeks, may undergo laboratory testing [155]. Brain imaging studies may be considered to rule out alternative etiologies or for research purposes [156]. Neuropsychological examination may also be recommended for patients experiencing cognitive symptoms affecting occupational and emotional well-being and social functioning. Attending physicians should remain vigilant regarding the psychological impact of COVID-19 and be prepared to conduct psychosocial evaluations, with referrals to psychiatrists as needed [157].

The early diagnosis of COVID-19 and assessment of its severity and long-term effect have been facilitated using machine learning (ML), Deep Learning (DL), and artificial intelligence (AI)-based diagnostic methods, either individually or in combination; see Figure 3. These methods rely on diverse patient data types, including imaging data such as chest X-ray, CT, magnetic resonance imaging (MRI), ultrasound, and non-imaging data like PCR, clinical records, epidemiological data, blood tests, and laboratory results. Additionally, remote video diagnostics and consultations are increasingly available in clinics and hospitals, with future advancements in AI and Natural Language Processing (NLP)-based technologies potentially leading to the development of remote video diagnostic programs, which could replace initial hospital visits for COVID-19 patients [158,159]. These approaches could be used for the early prediction of long COVID.

Pfaff et al. employed AI and health data from the National COVID Cohort Collaborators to devise machine learning models for identifying patients with long COVID and those necessitating specialized care. Their study demonstrated high accuracy in detecting potential long COVID cases, with key factors including outpatient clinic visits for long COVID, patient age, dyspnea, and concurrent diagnoses and medications. These findings highlight the significance of referring identified individuals to specialized clinics and streamlining patient recruitment for clinical trials, with the potential for model refinement based on emerging data sources [160]. Moreover, the utilization of machine learning models has yielded encouraging outcomes in forecasting long-term neurological consequences. Albaqer et al. [161] studied the application of the Random Forest model and achieved an accuracy rate of 85%, a sensitivity of 80%, and a specificity of 90% in identifying patients susceptible to developing neurological sequelae. These results underscore the importance of continuous monitoring and follow-up care for individuals recovering from COVID-19, especially regarding potential neurological complications. Integrating machine learning-based outcome prediction serves as a valuable asset for early intervention and tailored treatment approaches, aiming to enhance patient care and inform clinical decision-making.

## 9. Management and Treatment Strategies

### 9.1. Overview of Current Approaches to Managing Long COVID

There is an association between vaccination and a reduction in long COVID symptoms, with individuals receiving two doses of the COVID-19 vaccine showing lower odds of experiencing such symptoms compared to those who are unvaccinated [162]. Moreover, research indicates that individuals who receive two doses of the COVID-19 vaccine at least two weeks before infection may experience a 41% decrease in the risk of long COVID symptoms 12 weeks later [70]. Furthermore, the number of doses of the BNT162b2 vaccine appears to correlate with a reduced risk of long COVID, with meta-analysis confirming a 29% lower incidence of long COVID among vaccinated individuals compared to the unvaccinated [163,164]. However, despite the apparent effectiveness of vaccination in preventing long COVID, particularly with two doses, there remains a lack of comparative studies between different vaccine types in populations with long COVID. Thus, further research is warranted to assess the comparative efficacy of various vaccines in preventing and controlling long COVID symptoms [165].

To reduce the severity of COVID-19 and decrease the risk of hospitalization and death, the Food and Drug Administration (FDA) has approved various antiviral medications for mild and moderate cases of COVID-19. These medications include nirmatrelvir with ritonavir, remdesivir, and molnupiravir. Furthermore, the FDA has granted emergency use authorization for the prescription of immunomodulatory drugs, such as baricitinib and tocilizumab, for patients hospitalized with SARS-CoV-2 infection [166,167]. Long COVID manifests a wide range of persistent symptoms, necessitating a comprehensive and individualized treatment approach focused on symptom management and identification of treatable complications across various symptom clusters. Multidisciplinary rehabilitation incorporating ongoing symptom monitoring is recommended to enhance patients’ quality of life through tailored clinical interventions, physical rehabilitation, mental health support, and social assistance. Fatigue emerges as a predominant symptom, requiring prioritized attention in treatment approaches. Effective management strategies encompass pulmonary and cardiac rehabilitation, cognitive behavioral therapy, graded exercise therapy, strengthening exercises, and mindfulness training [168].

### 9.2. Pharmacological Management of Long COVID

In addressing chronic cough in long COVID, treatment strategies are guided by available therapies and established guidelines. Options such as oral corticosteroids, honey, opioid-derived antitussives, gabapentin, and pregabalin are viable for managing persistent cough. Integrating speech and language therapy into a multimodal approach to therapy and recovery can complement other aspects of pulmonary rehabilitation for individuals with long COVID. However, evidence is insufficient to support or discourage specific medical interventions for respiratory symptoms such as dyspnea and cough in long COVID patients. Similarly, there is insufficient evidence to support treatments for smell or taste disorders resulting from COVID-19 [169,170,171,172].

National Institute of Health (NIH) guidelines recommend continuing anticoagulants or antithrombotic medications in patients with underlying medical conditions. Routine thrombosis screening is not advised for asymptomatic patients, and preventive use of these drugs in non-hospitalized COVID-19 patients is not recommended [173]. Therefore, the need for thrombosis prevention in long COVID patients should be assessed based on their overall thrombosis risk and potential for bleeding, considering underlying health conditions [173].

Olfactory training may be suggested for patients with smell loss due to COVID-19, although further research is needed to evaluate other treatments. Phosphodiesterase inhibitors, insulin, and corticosteroids show promise as potential treatments for managing loss of smell and taste in COVID-19, given their neuroprotective, anti-inflammatory, or depolarizing properties according to current evidence [174]. Management of fatigue, headache, and cognitive impairment associated with long COVID warrants further investigation, with symptomatic treatments proposed as potential complements to drug therapy. Caution is advised with certain medications, and lifestyle adjustments are recommended alongside pharmacological interventions. Further evidence is required to confirm the efficacy of selective serotonin reuptake inhibitors in managing major depressive disorder in long COVID patients [143].

### 9.3. Non-Pharmacological Management and Rehabilitation

Fatigue emerges as a predominant symptom in long COVID, requiring prioritized attention in treatment approaches. Effective management strategies include pulmonary and cardiac rehabilitation, cognitive behavioral therapy, graded exercise therapy, strengthening exercises, and mindfulness training. Pacing, optimizing sleep hygiene, re-establishing self-care routines, and setting personalized activity thresholds are valuable interventions for fatigue management. Additionally, alternative approaches such as Chinese herbal formulation, molecular hydrogen (H2) inhalation, and hyperbaric oxygen therapy (HBOT) are used to improve fatigue in long COVID cases [175].

For individuals experiencing a chronic cough, integrating speech and language therapy into a multimodal approach can complement pulmonary rehabilitation efforts. Tailored respiratory rehabilitation may benefit those with persistent long COVID symptoms, especially those with prior ICU treatment or older adults [168].

Regular monitoring and classification of patients with neurological disabilities resulting from long COVID are essential. Persistent physical fatigue and breathlessness are common among these patients, regardless of race, age, gender, or socioeconomic status. Consequently, these patients require regular follow-up and management according to prescribed guidelines [176].

## 10. Conclusions

The prevailing trend of the presented results may indicate a strong relationship between the appearance and persistence of long COVID symptoms and the various comorbidities, together or individually. It has become known that not all acute SARS-CoV-2 (COVID-19) or severe infections are necessarily accompanied by long COVID. The orientation factors include the patient’s associated comorbidities, the patient’s gender, and the patient’s genetic predisposition. Understanding the role of autoantibodies in disease progression, thrombosis, and critical illness may lead to targeted therapeutic interventions and early detection strategies to mitigate the long-term impact of COVID-19 on individuals’ health outcomes. Exploring cardiovascular complications associated with long COVID highlights the complex interplay between viral infection, immune dysregulation, and pre-existing cardiovascular conditions. This underscores the need for a multifaceted approach to understanding and managing the long-term cardiovascular effects of COVID-19. Treatment involves a multidisciplinary approach focusing on symptom management, rehabilitation, and mental health support. Addressing the psychosocial impact is vital, alongside recognizing obesity and metabolic syndrome as risk factors. Despite diagnostic challenges, ongoing research and technological advancements offer hope for early detection and intervention. Overall, a holistic and collaborative approach is essential for optimizing patient outcomes and quality of life.

## 11. Challenges and Future Directions

Various AI-based applications are enhancing patient management by accurately predicting their conditions and hospitalization needs. AI’s ability to detect COVID-19 through chest X-rays enables early diagnosis and could potentially replace RT-PCR testing. Moreover, AI has been instrumental in forecasting COVID-19 outbreaks and conducting sentiment analysis of public opinions, thereby identifying misinformation and raising awareness about the pandemic. Additionally, ML, DL, and AI-based classification methods are continually being refined to improve the accuracy of COVID-19 detection and could be applied for the accurate detection of long COVID.

## Figures and Tables

**Figure 1 biomolecules-14-00835-f001:**
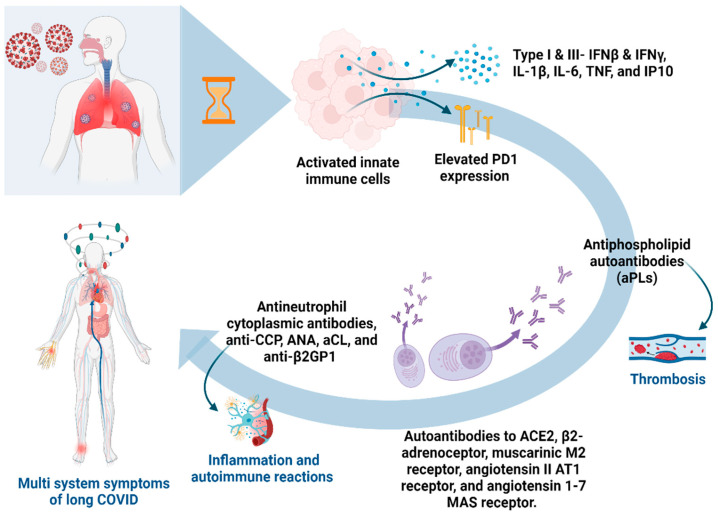
Comprehensive overview of immunological and autoantibodies responses in long COVID patients. Programmed cell death protein 1 (PD1); interferon (INF); interleukin (IL); tumor necrosis factor (TNF); induced protein 10 (IP10); angiotensin converting enzyme 2 (ACE2); cyclic citrullinated peptide (anti-CCP); antinuclear antibody (ANA); anti-cardiolipin (aCL), and anti-beta-2-glycoprotein-1 (anti-β2GP1).

**Figure 2 biomolecules-14-00835-f002:**
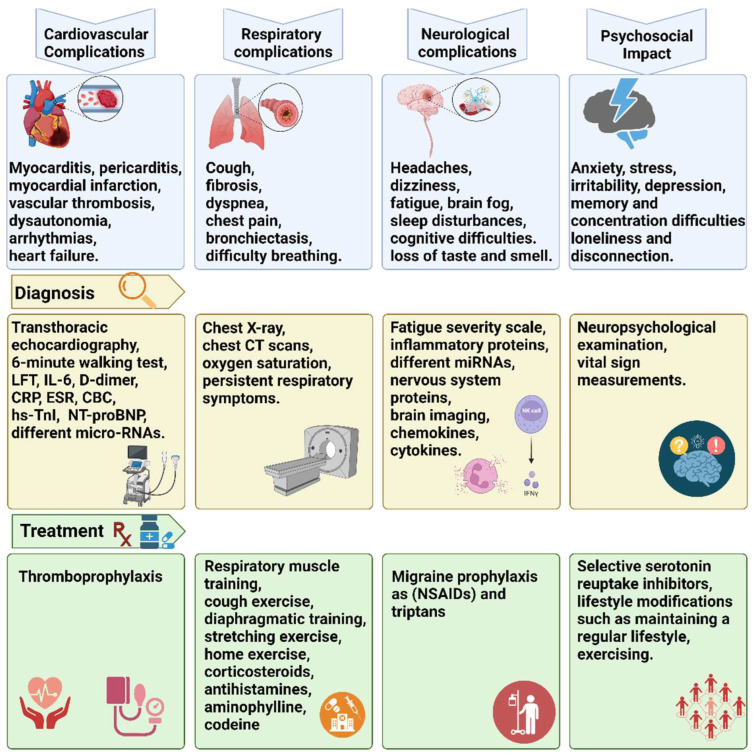
Comprehensive overview of complications, diagnostics, and treatment strategies for long COVID. This illustration provides detailed insights into the multifaceted approach to patient care. It highlights various complications, including cardiovascular, respiratory, neurological, and psychosocial impacts. Diagnostic methods for each complication type are outlined, along with potential treatment options. Liver function test (LFT); interleukin-6 (IL-6); C-reactive protein (CRP); erythrocyte sedimentation rate (ESR); complete blood count (CBC); high-sensitivity cardiac troponin I (hs-TnI); n-terminal brain-type natriuretic peptide (NT-proBNP); micro-RNAs (mi-RNAs); computed tomography (CT); and non-steroidal anti-inflammatory drugs (NSAIDs).

**Figure 3 biomolecules-14-00835-f003:**
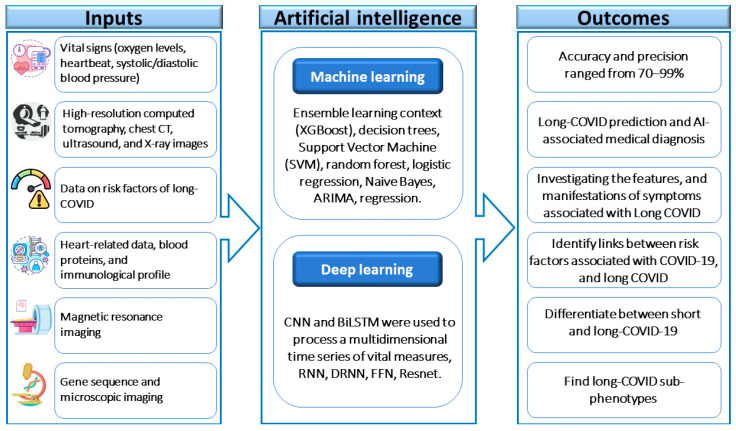
The framework of using artificial intelligence applications in the early diagnosis of long COVID.

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
