# Peer review of "Interplay between Comorbidities and Long COVID: Challenges and Multidisciplinary Approaches"

_biomolecules, 2024, doi:10.3390/biom14070835_

Round 1
Reviewer 1 Report
Comments and Suggestions for Authors
This is a very interesting review; determining the relationship between comorbidities and long COVID is of utmost importance for optimizing the management and outcomes of long COVID patients. I only have minor concerns.
Firstly, the title is not grammatically correct; please improve it. Throughout the manuscript, the authors overlooked some spelling errors, for example, in line 135 'dearth' should be 'death', and some others that involve the addition of articles. On the other hand, the authors should briefly describe the identity of specific autoantibodies associated with Long COVID and discuss in detail the mechanisms by which these autoantibodies might contribute to prolonged symptoms
Also, the authors should explore the relationship between molecular patterns of comorbidities and long COVID. Finally, starting from section 5.2 onwards, the authors repeatedly define long COVID in most subsequent sections, making the reading very repetitive.
Author Response
This is a very interesting review; determining the relationship between comorbidities and long COVID is of utmost importance for optimizing the management and outcomes of long COVID patients. I only have minor concerns.
- Firstly, the title is not grammatically correct; please improve it.
Reply: Thank you for your attention to this matter. The suggested new title is: “The Intrinsic Roles of Comorbidities in Long COVID”
- Throughout the manuscript, the authors overlooked some spelling errors, for example, in line 135 'dearth' should be 'death', and some others that involve the addition of articles.
Reply: Thank you for your comment. The entire manuscript has been reviewed and edited again, and these and other spelling and grammar errors have been corrected.
- On the other hand, the authors should briefly describe the identity of specific autoantibodies associated with Long COVID and discuss in detail the mechanisms by which these autoantibodies might contribute to prolonged symptoms
Reply: Thank you for your guidance, a paragraph illustrating the Specific Autoantibodies, and potential mechanism of their contribution to the symptoms was added in line 162
- Also, the authors should explore the relationship between molecular patterns of comorbidities and long COVID.
Reply: The related details were added to the section of introduction line 104
- Finally, starting from section 5.2 onwards, the authors repeatedly define long COVID in most subsequent sections, making the reading very repetitive.
Reply: Thank you for bringing this to our attention. All repeated definitions of long COVID have been removed.
Reviewer 2 Report
Comments and Suggestions for Authors
Congratulations to the authors for clearly, thoroughly and comprehensively describing the extensive published data. Despite, its appear quite long, the review is very didactic and precise. The sections are well organized and the figures well done. Chapter 8 is particularly interesting, highlighting the applications of AI in clinical practice.
Author Response
Congratulations to the authors for clearly, thoroughly and comprehensively describing the extensive published data. Despite, it appears quite long, the review is very didactic and precise. The sections are well organized and the figures well done. Chapter 8 is particularly interesting, highlighting the applications of AI in clinical practice.
Reply: Thank you for your positive feedback. We are pleased that you found our review clear, thorough, and comprehensive.
Reviewer 3 Report
Comments and Suggestions for Authors
Dear authors,
I suggest that you should look deeper into the methodology to understand the different findings in your results as compared to other studies regarding the number of symptoms during acute and chronic condition and pre/existing comorbidities and number of acute and chronic symptoms. The interview you used to collect the data maybe have a major impact to the results, while many studies usually collected questionnaires online or by paper
Another issue is that correlation analysis missess signifancies adjusted to the Bonferroni test. I suggest to perform regression analysis if possible.
Author Response
Dear authors,
- I suggest that you should look deeper into the methodology to understand the different findings in your results as compared to other studies regarding the number of symptoms during acute and chronic condition and pre/existing comorbidities and number of acute and chronic symptoms. The interview you used to collect the data maybe have a major impact to the results, while many studies usually collected questionnaires online or by paper
Reply: Thank you for your comment. The manuscript has been revised, and more detailed sections have been included.
- Another issue is that correlation analysis misses signifancies adjusted to the Bonferroni test. I suggest to perform regression analysis if possible.
Reply: Thank you for your observation. However, this review article does not include any statistical analysis.
Round 2
Reviewer 3 Report
Comments and Suggestions for Authors
Dear authors,
I apologize for sending "wrong" comments in the previous review process. It was meant for another article.
The major comment is regarding the title and aims. From the title and aims the reader expects a focus on co-morbidities and their impact on the postcovid-19 condition. However, only paragraphs 6 and 7 correspond to the purpose, while others analyse consequences or mechanisms of covid infection on different body systems. Please, adjust the content to the title and aims. In such case, shorten the review and keep the focus on co-morbidities and their impact. Otherwise, it is difficult to follow the "red line".
I would also appreciate to see more evidence on treatment and rehabilitation, maybe depending on the previous comorbidities, if is possible. You have mentioned approaches for treatment but I would like to read more about evidence-based treatments known today.
With best regards.
Comments on the Quality of English Language
Some minor spelling mistakes, for ex. "commodities" line 807.
Author Response
Dear authors,
I apologize for sending "wrong" comments in the previous review process. It was meant for another article.
The major comment is regarding the title and aims. From the title and aims the reader expects a focus on co-morbidities and their impact on the postcovid-19 condition. However, only paragraphs 6 and 7 correspond to the purpose, while others analyze consequences or mechanisms of covid infection on different body systems. Please, adjust the content to the title and aims. In such case, shorten the review and keep the focus on co-morbidities and their impact. Otherwise, it is difficult to follow the "red line".
I would also appreciate to see more evidence on treatment and rehabilitation, maybe depending on the previous comorbidities, it is possible. You have mentioned approaches for treatment but I would like to read more about evidence-based treatments known today.
With best regards.
Comments on the Quality of English Language
Some minor spelling mistakes, for ex. "commodities" line 807.
Thank you for your valuable feedback and suggestions. We apologize for any confusion caused by the initial comments. In response to your major comment, we have made several adjustments to the manuscript to ensure it aligns more closely with the title and aims. Therefore, we have revised the title to better reflect the content and focus of the manuscript. We have rechecked the manuscript to maintain a clear focus on co-morbidities and their effects. Non-relevant sections have been shortened or removed to ensure the "red line" of the manuscript is coherent and easy to follow.
Evidence on Treatment and Rehabilitation: We have expanded the sections on treatment and rehabilitation, incorporating more evidence-based strategies. The treatment section has been divided into three parts: an overview, pharmacological treatment, and non-pharmacological treatment and rehabilitation. This organization highlights the effectiveness of various approaches more clearly than before.
We believe these changes have significantly improved the manuscript, making it more focused and informative. Thank you again for your insightful comments.
Thank you for your feedback on the quality of the English language. We have conducted thorough proofreading to address and correct the identified issues